# The Study on Internal Flow Characteristics of Disc Filter under Different Working Condition

Yanbing Chi [1], Peiling Yang [1,*], Zixuan Ma [1], Haiying Wang [2], Yuxuan Liu [1], Bingbing Jiang [1] and Zongguang Hu [1]

[1] College of Water Resources and Civil Engineering, China Agricultural University, Beijing 100083, China; chiice@163.com (Y.C.); mazixuan@163.com (Z.M.); Yuxuan@126.com (Y.L.); bingbing@126.com (B.J.); Zongguang@126.com (Z.H.)

[2] Zhangjiakou Water Conservancy Development Planning Center, Zhangjiakou 075061, China; Haiying@163.com

[*] Correspondence: yangpeiling@126.com; Tel./Fax: +86-10-6273-7866

**Abstract:** A disc filter (DF) is an important component in a micro irrigation system. However, it has a high head loss and low filtration efficiency, which can lead to the inoperability of micro irrigation systems. To improve the filtration ability and to decrease the pressure loss of the irrigation system, it is necessary to internalize the hydraulic characteristics of DFs. In this study, the filter bed of a DF was divided into three parts, i.e., upper, middle, and lower, which were wrapped with a transparent film. The wrapped part was completely blocked. The purpose was to analyze the hydraulic characteristics of different clogged conditions in three types of filters under four types of flows. In addition, we attempted to simulate the filter operation process with computational fluid dynamics, based on two aspects—a macroscopic model and a simplified model. The results showed that the patterns of head loss among all of the DFs was consistent, and the macroscopic model that treated filter bed as a porous medium could express the measured results. The macroscopic models observed that there was a circular flow in the DF, and the flow velocity presented a symmetrical distribution in a horizontal direction. The middle of the filter element appeared in a high-pressure area and demonstrated the highest head loss, which may be the main flow area of the DF, and the inner flow characteristics of the DF were consistent under different conditions. The simplified models showed that the main flow area is near the filter bed in the inner DF, and the flow is tangent to the filter bed between 45 and 90 degrees in a horizontal direction. The uneven distribution of velocity and pressure on the filter bed might be necessary factors to impact filter efficiency.

**Keywords:** disc filter; filter bed; hydraulic characteristic; CFD; simplified structure

## 1. Introduction

The application of inferior water inhibits the promotion of water-saving irrigation technology. Filters are the key piece of equipment that influences the energy consumption of micro irrigation systems and uniformity in irrigation systems [1,2], which can remove impurities to avoid emitter clogging and to maintain the operability of micro irrigation systems. A disc filter (DF) is commonly used as a secondary filter in a micro irrigation system to remove inorganic and suspended organic particles in water. The disadvantages of DFs are their complex flow path structure, low filtration efficiency, long backwash time [3], and higher head loss than sand and mesh filters [4]. The complicated structure of DFs have had a significant impact on visualization research, and they were not conducive to product optimization and performance improvement. It is very necessary to gather detailed information about the inner flow characteristics and working mechanisms.

Most studies attempted to calculate head loss analytically, using methods such as dimensional analysis [1,5] and the Bayesian approach [6]. These models better predicted DF head loss by defining the relationship between the head loss and the structural parameters of the filter body, the physical parameters of the filter media, and the parameters of

the filtered liquid. In addition, several studies compared performance efficiency among different filters and illustrated the parameters of the DF, including the external/internal diameter, the groove interface/inclination, the number of laminations, and others that could impact the performance efficiency [3,7]. However, few researchers have revealed the internal flow mechanism of disc filter that is key to improving the filter efficiency and optimizing the structure of disc filter bed.

It is difficult to study internal flow because DFs have a complicated structure, with various laminations and channels. Computational fluid dynamics (CFD) is a tool that can better analyze the water path under complicated conditions, but there have been few studies on the application of CFD in DF. Most researchers have used CFD on drip emitters [8,9], water pumps [10], and other equipment in agricultural irrigation systems. The number of DF flow channels is huge, and the size of the flow channels gradually reduce from the outside to the inside, which is the key to improving the accuracy of the filter [11]. The flow patterns are usually evaluated according to Reynolds number; however, some researchers have shown that the flow patterns are complicated and are in complex and tiny channels [12,13]. It is difficult to judge the flow pattern under micro-scale flow using the Reynolds number, and the critical Reynolds number of micro-scale flow needs further study [14]. Therefore, it is difficult to analyze DF flow characteristics through numerical simulation. Simplifying the DF model is an effective way to solve the aforementioned problems. Some researchers have considered the whole DF as a porous medium [13,15], considering that the difference in the head loss was small between the numerical simulation and the measured test. Thus, reasonable simplification of the DF model is an effective way to solve the numerical DF simulation.

This paper focused on exploring the inner flow characteristics of DFs and their working mechanism through experiment and numerical simulation (CFD). The experiment analyzed the hydraulic characteristics of different clogged positions with different mesh filters at different flow rates. The numerical simulation attempted to visualize inner flow characteristics of DFs under a macroscopic 3D (three dimensional) model and a simplified 2D (two dimensional) model.

## 2. Materials and Methods

### 2.1. Working Setup

The field experiment was conducted at the experimental station of China Agricultural University (116°41′2.31″ E, 39°41′6.93″ N) in Tongzhou, Beijing, China. The experimental platform is shown in Figure 1. The workflow was defined as follows: (1) The groundwater was stored in the sedimentation tank. (2) The water that came from the sedimentation tank was pumped into the pumping tank using a submersible pump. (3) The difference of the pressure gauge showed the head loss of the tested DF. The quality of water is shown in detail in Table 1. The specific DF parameters are listed in Table 2.

### 2.2. Experimental Design

The platform used for the freshwater filtering test was shown on Figure 1. The test involved three types of DFs and three kinds of flows. Every filter element was equally divided into three parts by disc number, the upper part was UP, the middle part was MID, and the lower part was DOW. Each part was wrapped with a plastic film, which was assumed to be a complete blockage, as shown on Figure 1. The abbreviated number was $KD1_{UP}$, $KD_{1MID}$, $KD_{1DOW}$, $KD_{2UP}$, $KD_{2MID}$, $KD_{2DOW}$, $KD_{3UP}$, $KD_{3MID}$, $KD_{3DOW}$, and CK. CK is a freshwater filtering test without a filter bed, and $KD_1$, $KD_2$, and $KD_3$ is the head loss of different DFs not wrapped with plastic film. The flow was measured using a electromagnetic flowmeter (China, DN100, 4 V, 0150 ± 0.1 m$^3$/s), and the pressure was measured by a pressure gauge (ElECALL, 0–6 ± 0.1 KPa). Head loss was evaluated by the pressure difference before and after DF, and it was analyzed at different flows (5, 10, 20, 25, 30, and 35 m$^3$/h) of freshwater. $T_{KD_{iq}}$ indicates the proportion of head loss

between DFs wrapped in plastic film (UP, MID and DOW) and DFs not wrapped in a plastic film. The calculation is shown in Equation (1).

$$T_{KD_{iq}} = \frac{(KD_{iq} - KD_i)}{KD_i} \qquad (1)$$

*q* was UP, MID, and DOW; $KD_i$ indicates the head loss of the whole filter, and *i* was 1, 2, and 3;

The flowmeter and pressure gauge were set and calibrated by the manufacturer before the formal experiment. From Figure 1, the globe valve was used to adjust the flow to achieve the experimental flow. The pressure was recorded after the flowmeter was steady. The difference between front and rear pressure is the DF head loss.

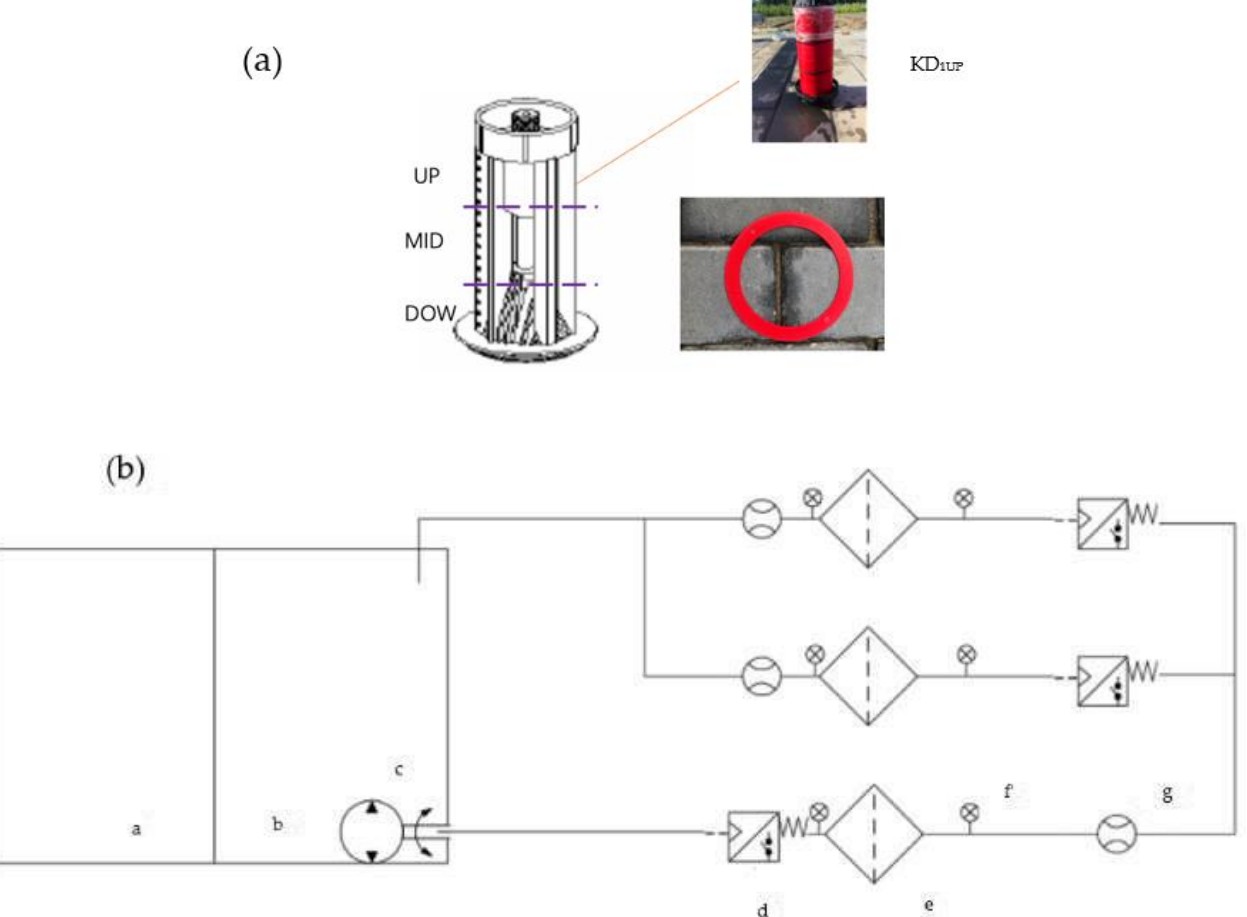

**Figure 1.** (**a**) Testing disc filter and (**b**) freshwater testing platform. a sedimentation tank, b pumping tank, c submersible pump, d globe valve, e disc filter, f pressure gauge, and g flowmeter.

**Table 1.** The quality of the water in the clean water test.

| Nov. | Test Items | Value |
|:---:|:---:|:---:|
| 1 | pH | 7.8 |
| 2 | Total suspended solid (TSS) mg/L | 7.7 |
| 3 | Total hardness mg/L | 983 |
| 4 | Chemical Oxygen Demand (COD) mg/L | 0.74 |
| 5 | Chloride mg/L | 167 |

**Table 2.** The DF parameters.

| Nov | Filter Accuracy | Disc Number | Weight (g) | Rated Pressure (MPa) | Rated Discharge (m³/h) | Groove Number |
|-----|-----------------|-------------|------------|----------------------|------------------------|---------------|
| $KD_1$ | 80 | 229 | 4.02 | 0.1 | 30 | 590 |
| $KD_2$ | 120 | 283 | 4.31 | 0.1 | 30 | 680 |
| $KD_3$ | 150 | 365 | 3.29 | 0.1 | 30 | 710 |

$KD_1$, $KD_2$, and $KD_3$ were the abbreviations for different filters; filter accuracy was the number of pores per square meter; disc number was the number of laminations on filter element.

### 2.3. Model Setting

Simplifying the DF model included two methods: one was to place filter element into porous media (3D), and the second was to transform the 3D model (DF) into a 2D model. To ensure the availability of the simulation results, the study uses the following process: This paper uses the head loss that was measured by the experiment to verify the simulated head loss. If the simulation results were consistent with the experimental results, the study would consider that the simplified model was reasonable [16]. Therefore, it is once again emphasized that this study only explored the feasibility of the simplified model, which could effectively improve DF manufacturing efficiency.

### 2.4. 3D Model Setting

#### 2.4.1. The Mesh Setup

The entire structure of the DF was stripped and simplified. The simplified model is shown in Figure A1, and the size is listed is Table A1. In order to connect the parts normally and to complete the data transmission, the surfaces or planes were that connected by the two parts were set to interface; thus, it could ensure that every part had the same grid scale when the partition types were different. As shown in Figure A1, the unstructured grid was used for the four parts. The mesh of every part is also shown in Figure A1.

#### 2.4.2. Calculation Model

In this paper, the flow pattern belongs to turbulent flow according to the Reynolds number calculation. However, this would produce certain friction between the flow and the wall, and the Reynolds number would then be in a state of change. The paper conducts the numerical simulation of the three models (standard k-ε, Re- normalization group k-ε, and realizable k-ε). All turbulent models adopt the default setting of fluent 17.2. The calculated method was the SIMPLE pressure correction segregated solver [17,18]. The residual was $10 \times 10^{-4}$. Comparative analysis found that the realizable k-ε model, which is more suitable for the internal flow process of the filter, was selected, as shown in Figure A3. $RMSE\text{rea} = 0.26$, $RMSE_{RNG} = 0.332$, and $RMSE\text{sta} = 0.341$. the calculation of $RMSE$ was completed as follows:

$$RMSE_a = \sqrt{\frac{1}{n}\sum_{i=1}^{n}(y_i - \hat{y}_i)^2} \tag{2}$$

$RMSE$ (root mean square error) was the difference between the measured head loss and the simulated head loss; a was the rea, RNG and sta represented realizable k-ε, RNG k-ε and standard k-ε, respectively; $y_i$ was the measured head loss of KD2 under different working conditions; $\hat{y}_i$ was the simulated head loss of realizable k-ε, RNG k-ε, and standard k-ε under different working conditions.

#### 2.4.3. Boundary Conditions and Solved Model

All boundary conditions were set according to the testing data. The boundary condition of inlet was velocity-inlet, and its value was calculated by the flow and inlet area. The obtained velocities were 0.445. 0.891, 1.782, 2.225, 2.673, and 3.115 m/s. The boundary condition of outlet was pressure-outlet, and the wall roughness was set to 0.5 based on the material properties of the DF. Meanwhile, the model set the flow along the axial direction of

the DF element to zero. The main setting parameters of porous media includes the viscous resistance coefficient ($C_1$) and internal resistance ($C_2$). The formula is as follows:

$$C_1 = \frac{1}{\alpha} \tag{3}$$

$$C_2 = \frac{3.5}{D} \frac{(1 - \varepsilon)}{\varepsilon} \tag{4}$$

$$\alpha = \frac{D^2}{150} \frac{\varepsilon^2}{(1 - \varepsilon)^2} \tag{5}$$

$$D \times \varphi = 15,000 \tag{6}$$

these formulas refer to fluent 17.2 manual [18]. D was average particle size, and the filter accuracy ($\varphi$) was 80, 120, 150, so the value of D could be set to 0.1875, 0.125, or 0.100 mm; the porosity ($\varepsilon$) was 0.4.

### 2.5. 2D Model Setting

The horizontal simple model depicted the top view of the DF. Although the horizontal simple model ignored the vertical flow, it could partially affect the distribution of velocity and pressure around the filter bed. The related model setting in Fluent 17.2 referred to the 3D model. The boundary conditions of all of the simple models were the outflow; the boundary conditions of inlet were the velocity. The inlet velocities were 1, 2, and 3 m/s. The related model setting in Fluent 17.2 referred to the 3D model.

The vertical simple model depicted the front view of the DF. The vertical simple model could not react to the actual situation, as the vertical simple model ignored the horizontal flow. Therefore, the back part of the vertical simple model had no practical significance. The front part illustrated the distribution of the vertical velocity, excluding the horizontal velocity. The outlet boundary conditions of all of the simple models were the outflow; the inlet boundary conditions were the velocity. The inlet velocities were 1, 2, and 3 m/s. The related model setting in Fluent 17.2 referred to the 3D model.

### 2.6. Analysis Method

In this study, we used ANSYS Fluent 17.2. All statistical analyses were performed using SPSS V26 for Mac (SPSS Inc., Chicago, IL, USA). Repeated measures of analysis of variance (ANOVA) were performed to test the differences in head loss and velocity at a significance level of 0.05.

## 3. Results

### 3.1. Filter Test

As expected, the Df head loss increased with the increasing flow velocity, and the higher filter accuracy could raise head loss among all of the treatments, as shown in Figure 2. The $KD_3$ (mesh number: 365) had the maximum head loss, which was significantly ($p < 0.05$) different with other velocities. Different blocking positions yielded different head loss values ($KD_{iMID} > KD_{iUP} > KD_{iDOW} > KDi$ (i = 1, 2, 3)). The $KD_{iMID}$ treatment resulted in the highest value among all of the treatments under the $KD_i$ filter; the gradient (the ratio of head loss to flow) between 20 and 25 $m^3$/h had the highest value among all of the $KD_{iMID}$ treatments form Figure 1.

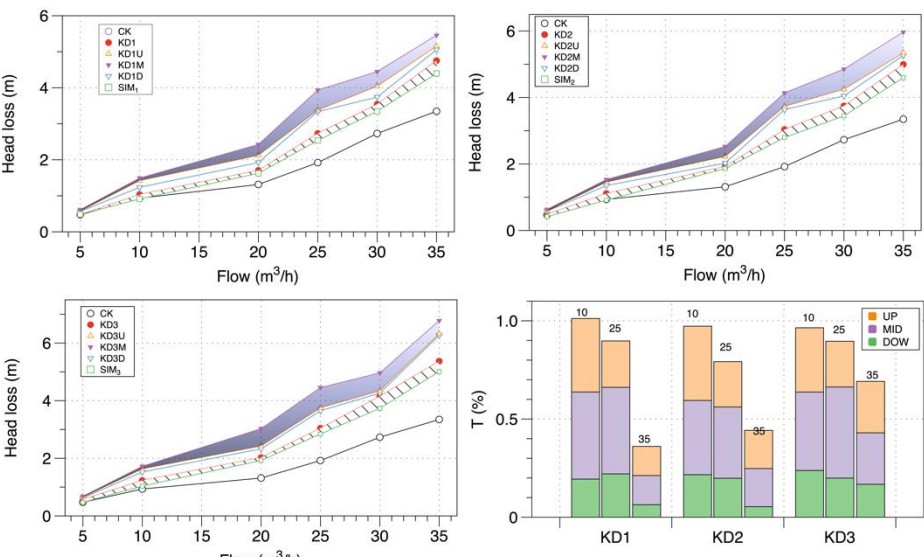

**Figure 2.** The pattern of head loss in different treatments. The gradient was the difference between KDM and KDU, and the pattern was the difference between $KD_i$ and $SIM_i$. $SIM_i$ was the head loss of the whole filter simulation, and i was the DF number.

### 3.2. Macroscopic Simulation

The results of the simulation had a small difference when compared to the measurements shown in Figure 3, and the simulated value was lower than the measured value. The difference between the measured and simulated results increased with the increasing flow; it was the highest value in which the flow rate was 35 $m^3$/h. As shown in Figure 3a, the $R^2$ correlation between the simulated value and measured value was 0.677, and the slop k was 1.0929. The comparison depicted that the simulated value was consistent with the measured value. From Figure 3b, the velocity around the filter bed was lower than it was in other areas, and there was water flow around the column, as show in Figure 3c. The velocity increased because of the filter bed base, and the velocity was lower around filter shell from Figure 3c. the higher eddy viscosity appeared on the filter bed base and the outlet from Figure 3c. The pressure of the filter at the front part was higher than the filter in the back part (Figure 4a). As shown in Figure 4b, the study considered the pressure increases around the filter element.

### 3.3. Horizontal Simple Simulation

The main flow moved, and the velocity in the horizontal simple model had a symmetrical distribution around the filters in all of the treatments; the distribution was not affected by the value of the inlet speed. The mainstream velocity increased with the increasing inlet speed from Figure A4. The speed near the wall was lower than that in the inside part (Figure 5). A significant velocity gradient was observed in the ranges of 45°–90° and 270°–315°, and the direction of the velocity was primarily along the filter element from Figure 5. The front of filter bed might have caused high head loss.

### 3.4. Vertical Simple Simulation

A primary flow appeared where the velocity was higher than the others, as shown in Figure 5. The fringe of the filter base appeared, and the speed increased rapidly, as shown in Figure 5. The phenomenon was not affected by the inlet flow condition from Figure A4. The velocity at the wall of the filter was smaller than that at the inside part (Figure 5), and this outcome was similar to that of the 3D and horizontal simulation. Our study considered MID as the main filter area because its velocity was more than the velocities in the UP and DOW parts. The main water flow moved toward the upper direction with the increasing inlet speed (Figure 5). These two parts might cause high local head loss, which increased

with the increasing inlet speed. The direction of the velocity in DOW was opposite to the UP and MID velocity (Figure 5).

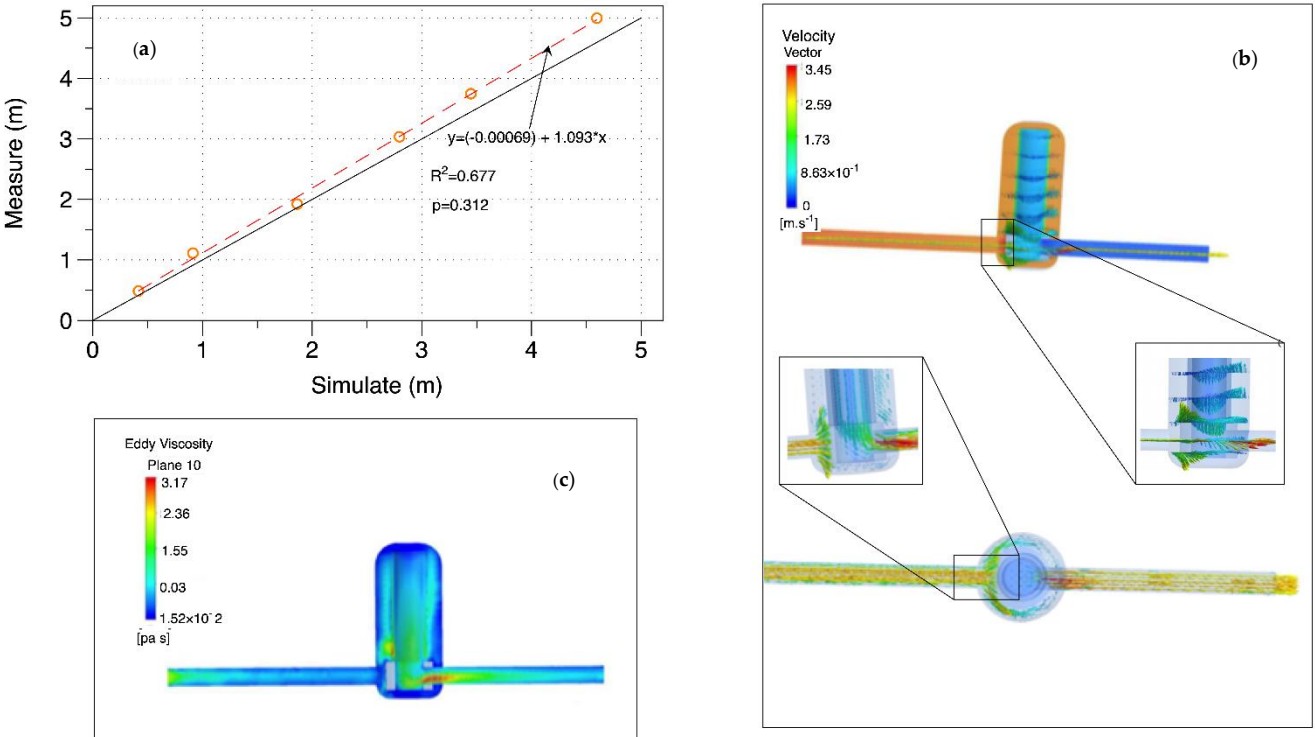

**Figure 3.** Results of 3D simplified model simulation. (**a**) Dotted line denotes linear regression fitting between the calculated and measured values. (**b**) Velocity distribution of KD2 (whole simulation) under 30 m$^3$/h. (**c**) Eddy viscosity of KD2 (whole simulation) under 30 m$^3$/h.

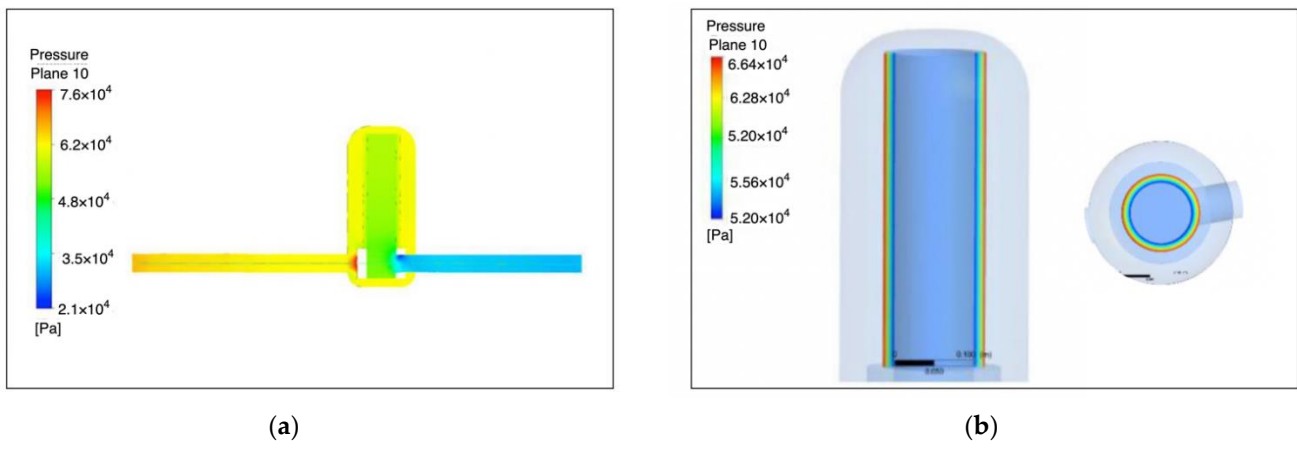

**Figure 4.** (**a**) The pressure nephogram of the whole water body; (**b**) the distribution of pressure in the filter bed.

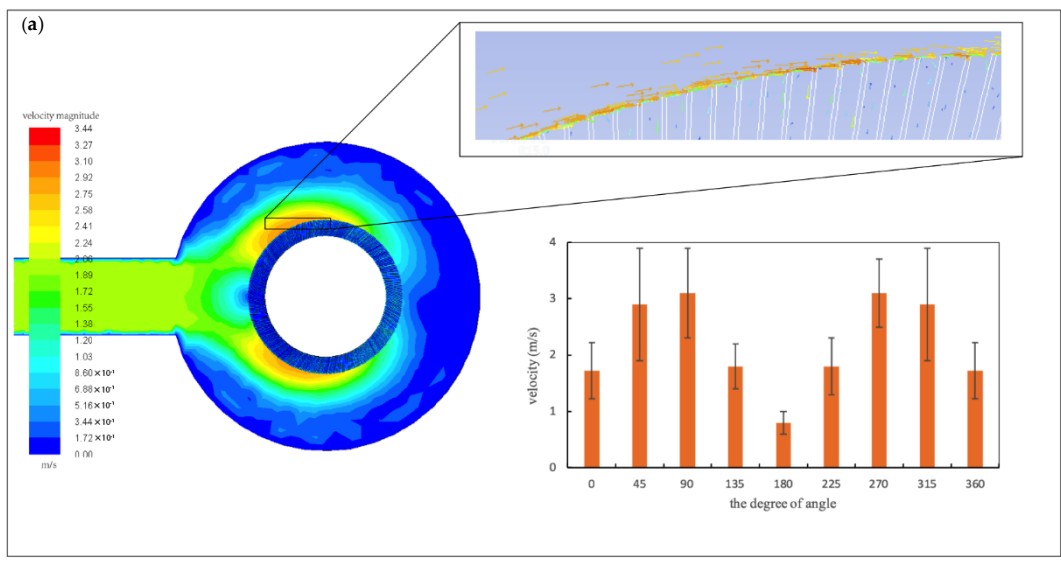

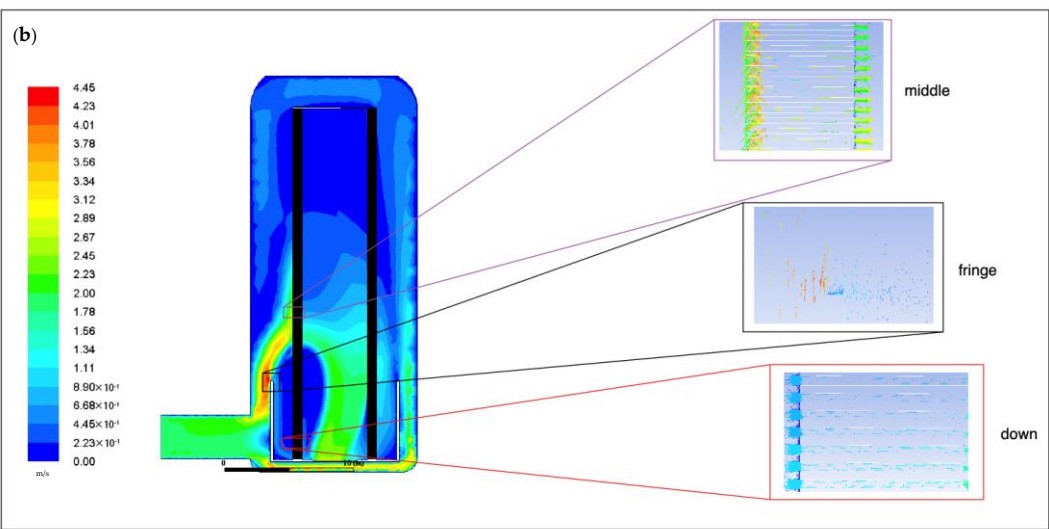

**Figure 5.** Local results of the simplified model. (**a**) The velocity distribution around the filters; the study randomly selected 10 points for each angle. The same letters denote "not significant ($p > 0.05$)"; the different single letters denote "significant ($p < 0.05$)"; the double letters mean "extremely significant ($p < 0.01$)". The angle formed by clockwise rotation with the entrance direction; (**b**) simulated results of the vertical simple model with 566 laminations at 2 m/s. "Middle/Down" denotes the random region in the middle/down part of the filter bed. "Fringe" denotes the region in the fringe of the filter base. b. Simulated results of the horizontal simple model with 340 grooves at 2 m/s, shown in the bottom right of the picture.

## 4. Discussion

Head loss is considered to be an important factor micro irrigation filtering systems. Despite the difference in research methods, several studies have considered the structure of filter media and working conditions, which have played an important role in energy loss. Bové et al., 2017, showed that a large number of meshes of sand media filters could affect the head loss of an underdrain. Wu et al., 2014, stated that the inlet/outlet inner diameter, inflow path depth, and inner diameter of a DF bed could be increased to increase the cross-sectional area of the flow path, decrease the velocity, and, thus, reduce the head loss. García Nieto et al., 2018, stated that the hybrid model could predict the pressure drop and that the flow surface velocity (input variable) was the most influential parameter in producing head loss. This study had some similar results that concluded that the inlet

velocity and the different mesh numbers of the DF were important factors that could affect head loss.

As shown in Figure 3a, our study considered that the simulated head loss was lower than the measured value; however, there was no significant difference between them. Therefore, the study considered a macroscopic simplified model, which was reasonable. Moreover, the measured results showed that KD$_{iMID}$ had the highest head loss in comparison to other treatments, and the results coincided with the macroscopic simulation. Our study considered that the assumed condition that the porous medium of the filter bed could be applied to DF simulation, but this assumption could have had an impact on the internal flow. The simulated value in the 3D model could not truly reflect the effect of the flow path transition on flow velocity and pressure. Thus, the simulated head loss was less than the measured head loss.

As the local head loss was easily caused by the structure and high speed, it was necessary to refine the internal structure of the DF. With regard to the second simplification, the simplified model placed the whole filter into the horizontal and vertical models. As shown in Figure A4, the assumed model showed that MID had the highest velocity, which was the main filter area. This is because some areas had deviations in the velocity direction. The important area was the filter base, which changed the direction of the flow and increased speed (Figure A4). As a result, MID had the highest head loss in the second simplified model. The results were similar to the results of the measured test and the macroscopic simulation. Therefore, the study assumed that the vertical simplified model could be applied to the simulation of the DF, but there were some problems. The filtration process involved the accumulation of water flowing from the bottom to the top sections and from the front to the back sections. The horizontal model ignored the vertical speed, so the influence of the accumulation from bottom to top on the horizontal velocity could not be evaluated. Moreover, the horizontal velocity was ignored; the model did simulate the circulating flow process from the bottom to the top. Therefore, research on reasonable construction models still needs to continue.

## 5. Conclusions

The fresh testing experiment showed that head loss patterns are consistent with the flow rate among all DF treatments. The experiment could better illustrate that the flow characteristics and working mechanism was similar among all DF treatments. Numerical simulation is feasible for the visualization of DF, as it could promote the development of a CFD numerical DF simulation. The velocity and pressure distribution of the DF was not impacted by the working conditions. The middle part of the DF was the main filter area that had the highest head loss, and the head loss was uniformly distributed on the filter bed. Only by effectively relieving the velocity and pressure in the middle of filter bed can the overall filtration efficiency and accuracy be improved.

**Author Contributions:** Y.C. and P.Y. designed the experiments; Y.C. and Z.M. conducted the experiments; Y.L. and B.J. contributed materials; Y.C. analyzed the data and wrote the paper; H.W. and Z.H. revised the paper. All authors have read and agreed to the published version of the manuscript.

**Funding:** This research was funded by the National Key Research and Development Project of China, grant number (No. 2019YFC0408703.) and the Key Research and Development Project in Hebei Province (No.1822700D).

**Institutional Review Board Statement:** Not applicable.

**Informed Consent Statement:** Not applicable.

**Data Availability Statement:** The data presented in this study are available on request from the corresponding author. The data are not publicly available due to the project is not finished.

**Conflicts of Interest:** The authors declare no conflict of interest.

## Appendix A

**Table A1.** The size of the 3D simplified model.

| Program | Size/mm |
|---|---|
| Diameter of inlet | 63.00 |
| Length of inlet | 750.00 |
| Length of outlet | 510.00 |
| External diameter of filter element | 130.00 |
| Inner diameter of filter element | 102.00 |
| Diameter of DF shell | 240.00 |
| Total high of DF | 520.00 |
| High of filter element | 360.00 |
| High of base | 105.00 |

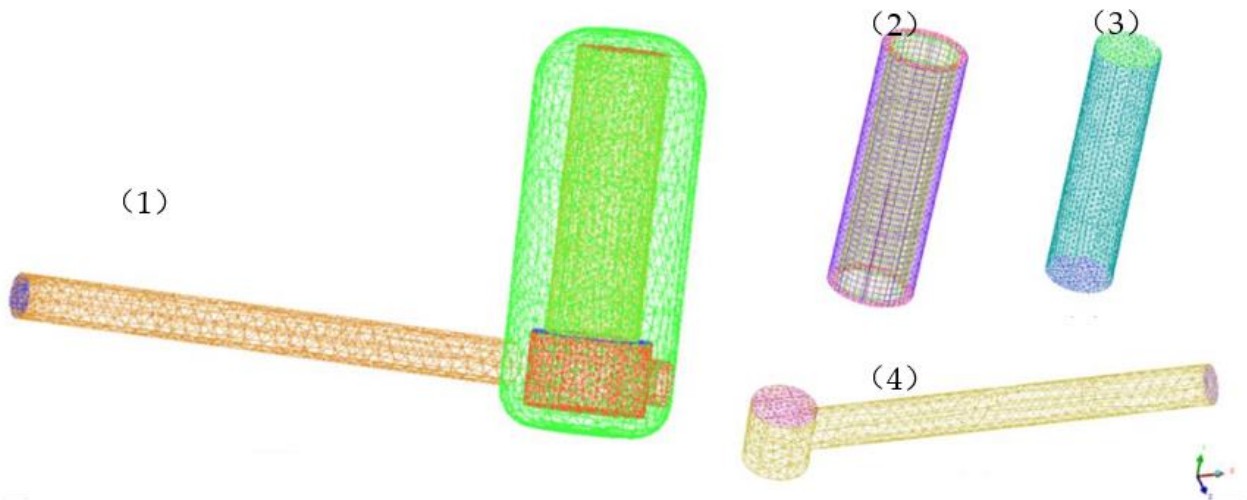

**Figure A1.** The 3D model of DF: (**1**) import body, (**2**) filter element, (**3**) inner body of filter bed, (**4**) outlet flow; model grid generation. the import body has 199166 nodes, the filter element has 5987 nodes, the inner body of the filter bed has 5145 nodes, and the outlet body has 7941 nodes.

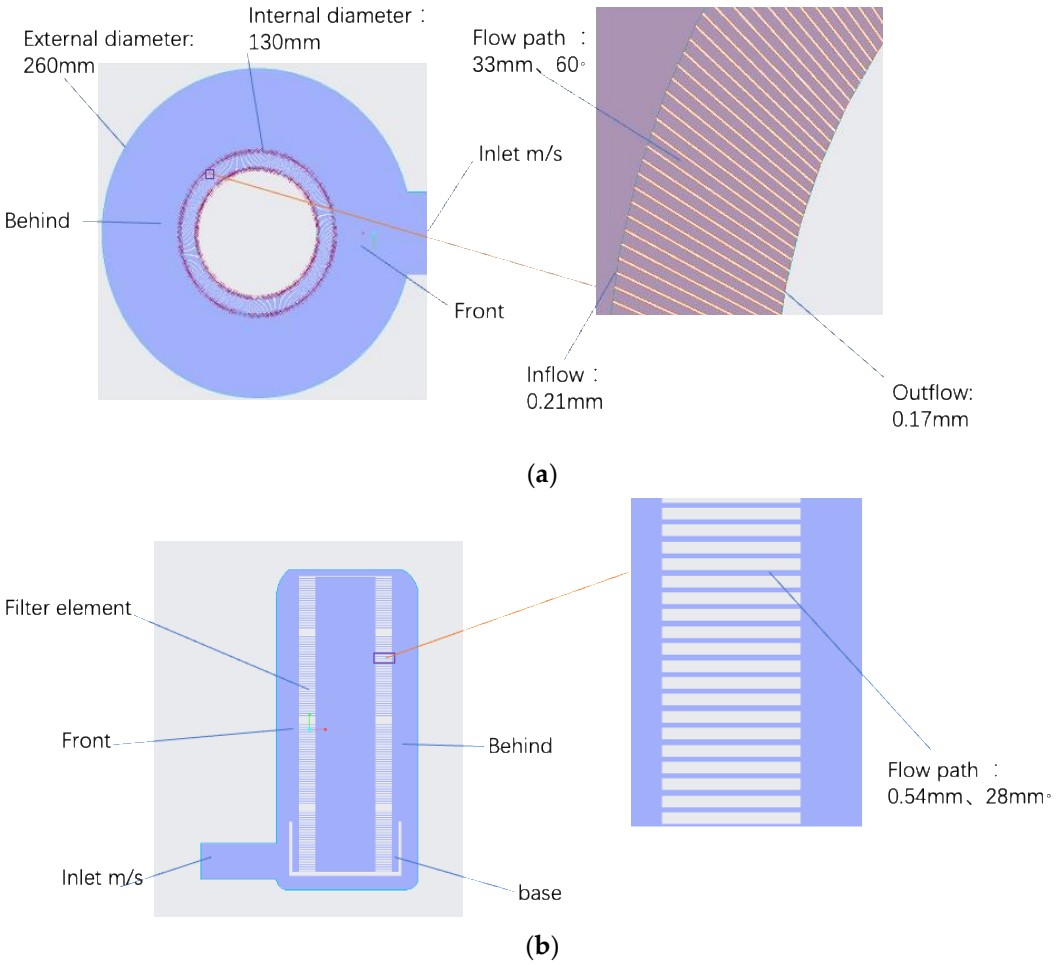

(**a**)

(**b**)

**Figure A2.** (**a**). 2D model denotes the horizontal part of the DF. The number of grooves was 340. The horizontal simple model had 393,886 nodes; (**b**). 2D model denotes the vertical part of DF. The number of flow paths was 566. The vertical simple model had 1,515,397 nodes.

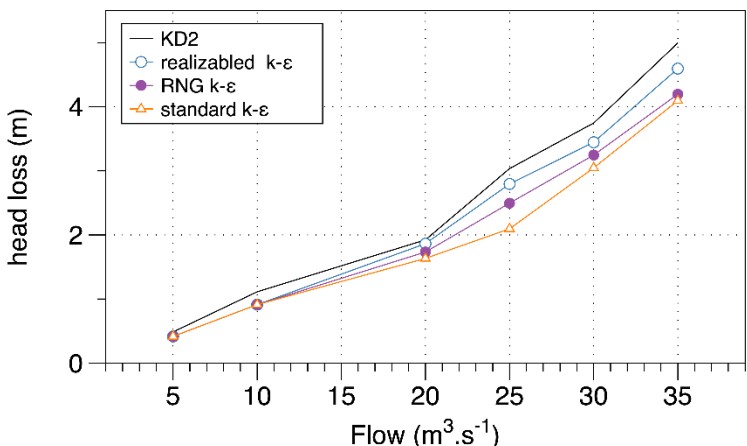

**Figure A3.** The simulated results among all turbulence models.

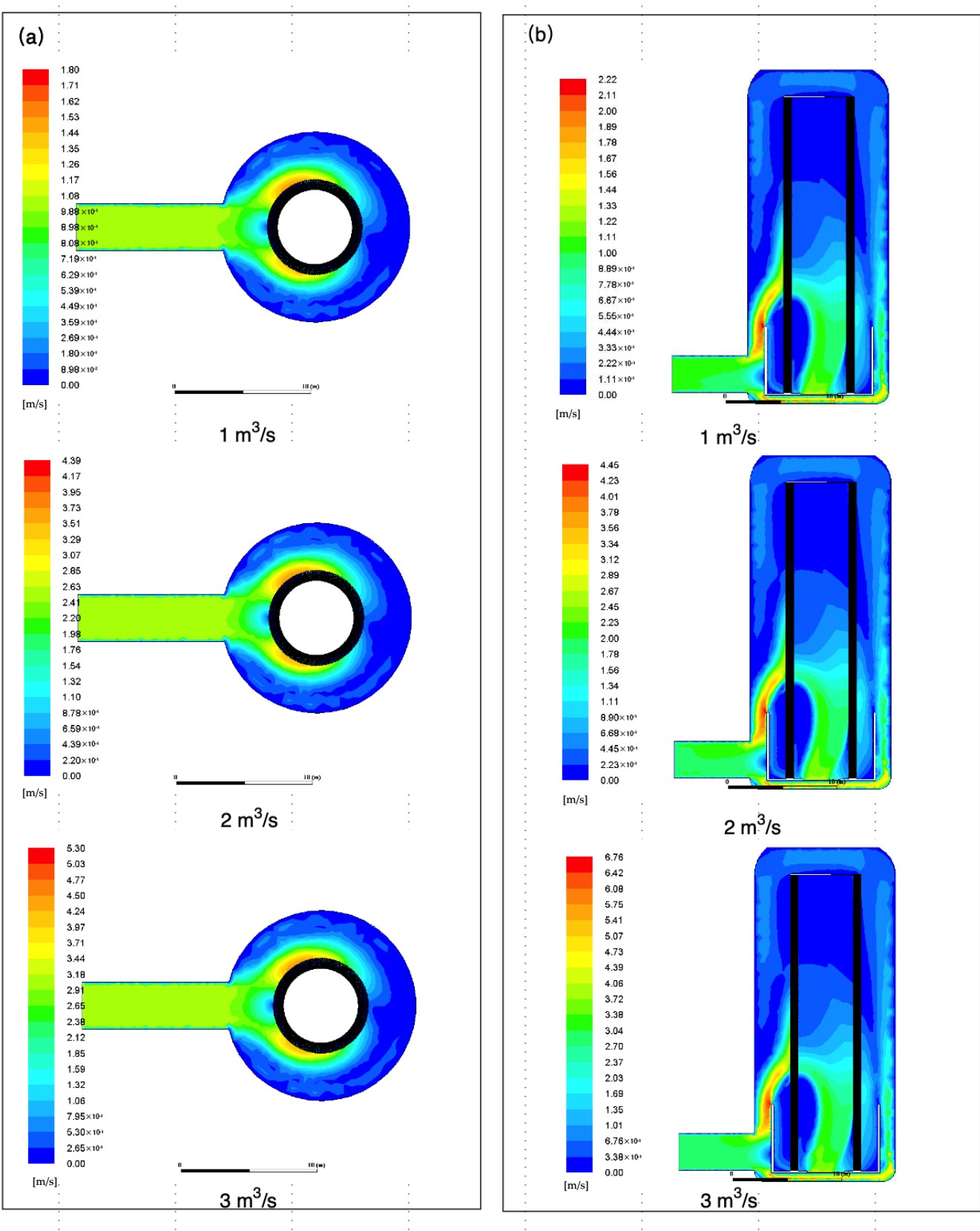

**Figure A4.** Velocity contours for different flow rate conditions: (**a**) Velocity pattern in the horizontal simplified model. (**b**) Velocity pattern in the vertical simplified model.

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
