# Peer review of "The Study on Internal Flow Characteristics of Disc Filter under Different Working Condition"

_applsci, doi:10.3390/app11167715_

Round 1
Reviewer 1 Report
The present paper focused on exploring inner flow characteristics of DF and working mechanisms through experiment and numerical simulation (CFD). The experiment analyzed the hydraulic characteristics of different clogged positions with different mesh filters at different flow rates, and verify the accuracy of numerical simulation; a macroscopic and a simplified model under CFD attempted to visualize inner flow characteristics of DF. The results showed that the patterns of head 17 loss among all DF was consistent, and the simplified 3D model could express the measured result.
The structure of the paper is properly organized. It is an interesting and well-written paper. The abstract is well written, the analysis seems accurate and clearly explained, the results look secure, and the conclusion is mature. The abstract should be concise. References need some attention and should be updated. Cite the following more references to improve the quality of the manuscript in support of CFD application:
https://www.tandfonline.com/doi/full/10.1080/02533839.2014.998165
https://link.springer.com/chapter/10.1007/978-981-16-0159-0_49
https://link.springer.com/chapter/10.1007/978-981-13-6416-7_57
The paper could be accepted after making the mentioned minor corrections.
Author Response
Thanks for your affirmation of the content of the article, your comments have greatly improved the quality of my paper. I have carefully read some references that you have recommended, these contents could better support the application of CFD. The related content had been added in the manuscript on Line138.
Reviewer 2 Report
The current manuscript presents an approach to assess the hydraulic characteristics of a disc filter (DF) under different working conditions. To this end, the authors used an experimental setup of freshwater testing as an example. A commercial software ANSYS is also used. However, it is not clear why ANSYS is used. Additionally, it is also not clear what kind of simulation model is implemented? The pressure losses in three parts of DF have been evaluated. In my opinion, there is no novelty in the methods. The interest could be in the particular application. The presented manuscript should be improved significantly to meet the scientific standards. I have added my suggestions in the pdf file. I propose to accept this manuscript after major revisions.

Author Response
Thanks for your comments, these could greatly improve the quality of my paper. We answer all the comments one by one, and the corresponding changes in the revised manuscript are marked in yellow.
